# Potential Anti-Rheumatoid Arthritis Activities and Mechanisms of *Ganoderma lucidum* Polysaccharides

**DOI:** 10.3390/molecules28062483

**Published:** 2023-03-08

**Authors:** Meng Meng, Jialin Yao, Yukun Zhang, Huijun Sun, Mozhen Liu

**Affiliations:** 1Department of Orthopaedics, First Affiliated Hospital, Dalian Medical University, Dalian 116011, China; 2Department of Clinical Pharmacology, College of Pharmacy, Dalian Medical University, Dalian 116041, China; 3Chongqing Key Laboratory of Development and Utilization of Genuine Medicinal Materials in Three Gorges Reservoir Area, Chongqing 404120, China

**Keywords:** rheumatoid arthritis, *Ganoderma lucidum* polysaccharides, anti-inflammation, immunomodulation, cartilage protection

## Abstract

Rheumatoid arthritis (RA) is a chronic and autoimmune disease characterized by inflammation, autoimmune dysfunction, and cartilage and bone destruction. In this review, we summarized the available reports on the protective effects of *Ganoderma lucidum* polysaccharides (GLP) on RA in terms of anti-inflammatory, immunomodulatory, anti-angiogenic and osteoprotective effects. Firstly, GLP inhibits RA synovial fibroblast (RASF) proliferation and migration, modulates pro- and anti-inflammatory cytokines and reduces synovial inflammation. Secondly, GLP regulates the proliferation and differentiation of antigen-presenting cells such as dendritic cells, inhibits phagocytosis by mononuclear macrophages and nature killer (NK) cells and regulates the ratio of M1, M2 and related inflammatory cytokines. In addition, GLP produced activities in balancing humoral and cellular immunity, such as regulating immunoglobulin production, modulating T and B lymphocyte proliferative responses and cytokine release, exhibiting immunomodulatory effects. Thirdly, GLP inhibits angiogenesis through the direct inhibition of vascular endothelial cell proliferation and induction of cell death and the indirect inhibition of vascular endothelial growth factor (VEGF) production in the cells. Finally, GLP can inhibit the production of matrix metalloproteinases and promote osteoblast formation, exerting protective effects on bone and articular cartilage. It is suggested that GLP may be a promising agent for the treatment of RA.

## 1. Introduction

Rheumatoid arthritis (RA) is a chronic, autoimmune inflammatory disease with a global prevalence of 0.5~1% and is more prevalent in females [1,2]. More than 2/3 of RA patients are characterized by the presence of autoantibodies, such as rheumatoid factor (RF) and disease-specific autoantibodies against citrullinated proteins, called anti-citrullinated protein/peptide antibodies (ACPA) [3,4]. The main clinical manifestation of RA is a chronic, symmetrical, peripheral type of polyarthritis inflammatory lesions [5]. If not treated properly, it can lead to irreversible joint damage and bone destruction, which can lead to permanent disability and seriously affect the patient’s physical function and quality of life [6]. The main extra-articular symptoms involve cardiovascular, ocular, neurological, and pulmonary, and sometimes also include abnormalities in blood, liver, and kidney function, which are usually found in long-term RA patients, and occasionally even lead to death [7].

Until now, the cause of RA remains unclear [8]. It is now believed that articular cartilage and bone are destroyed by inflammatory and immune mechanisms, which involve a variety of pro-inflammatory factor-secreting cells, including immune cells (e.g., mast cells, macrophages, monocytes, dendritic cells (DCs), T and B cells) and synovial cells (macrophage-like synovial cells and fibroblast-like synovial cells (FLS)) [9]. With the onset of RA, pro-inflammatory factors such as tumor necrosis factor (TNF)-α, interleukin (IL)-1β and IL-6 are first released by intra-articular immune cells and macrophage-like synovial cells to promote the infiltration of various immune cells such as adaptive immune cells, T cells and B cells into the synovium, exacerbating the secretion of pro-inflammatory factors and worsening RA by a vicious feedback cycle [10]. In this cycle, TNF-α, IL-1β and IL-6 activate FLS, leading to synovial proliferation, while activated fibroblast-like synoviocytes secrete matrix metalloproteinases (MMPs) and activate osteoclast production to result in bone erosion and cartilage degeneration [11]. TNF-α, IL-1β and IL-6 also stimulate the formation of new blood vessels and facilitates the transfer of inflammatory mediators and inflammatory cells [12,13].

The aims of the treatment of RA are reducing joint inflammation and pain, preventing progressive joint damage, and ultimately relieving or reducing disease activity [12]. In the past few decades, the mainstay of RA therapeutic agents has been disease-modifying anti-rheumatic drugs (DMARDs; e.g., methotrexate), non-steroidal anti-inflammatory drugs (NSAIDs; e.g., ibuprofen, aspirin, and naproxen), glucocorticoids, and biologics [14]. These drugs modulate the abnormal immune response by suppressing immune function in the progression of RA [15]. However, the results so far have been unsatisfactory, and the long-term use of these drugs has been associated with many adverse effects and limitations, such as infections and liver and kidney damage [16]. Because of the low adverse effects and low toxicity of traditional herbal medicines (TCM), they have attracted the interest of researchers [17]. *Ganoderma lucidum* (Leyss.ex Fr.) Karst is a wonder herb with immunomodulatory properties that is widely used in China, Japan, Korea, and other oriental countries [18,19]. It has been found that *G. lucidum* can alleviate the pain of RA patients without side effects [20,21]. In particular, *G. lucidum* polysaccharide (GLP) is one of the main active components, which has various pharmacological effects such as immunomodulation, antioxidation, hepatoprotection, anti-proliferation and anti-angiogenesis, which would be a potential therapeutic agent for RA [22,23].

Adjuvant-induced arthritis (AIA) and collagen-induced arthritis (CIA) are the most widely used animal models of RA, reflecting the clinical features of RAas manifested by inflammatory joint swelling, synovial tissue hyperplasia, and cartilage and bone destruction [24,25]. In addition, biological sample models of human origin can also be used for RA research and drug screening, such as FLS, human monocyte-derived DCs and peripheral blood mononuclear cells [26]. Based on the previous studies, this paper reviewed the cellular and molecular mechanisms underlying the potential anti-RA effects of GLP and focused on recent advances in the anti-inflammatory, immunomodulatory, anti-angiogenic and osteoprotective effects of GLP. Moreover, the protective effects of GLP on organ injury caused by immunological disorders were also supplemented to provide a reference for the use of GLP in the treatment of RA.

## 2. Methods

All the information about this natural product has been obtained through searching journals, books, and theses collected via libraries or electronic databases, including PubMed, Elsevier, Google Scholar, and Springer.

## 3. Inhibition of GLP on Synovial Joint Inflammation

### 3.1. Behavior of Synoviocytes

In RA, there is persistent synovitis, the main pathological manifestation of which is the activation and proliferation of macrophage-like synovial cells and FLS [27]. These cells are important non-immune cells located mainly in the inner layer of the synovium [28]. Synovial cell proliferation and the formation of new blood vessels lead to excessive growth of fibrous tissue. The formation of pannus invades cartilage and subchondral bone, resulting in erosion and joint deformity [29]. The pathological mechanism of RA is illustrated in Figure 1. In both AIA and CIA rats, significant allodynia, edema, hyperemia, immune cell infiltration, synovial tissue proliferation, and erosions of joint cartilage can be observed in the ankle joints [30,31]. Intraperitoneal injection of the *G. lucidum* suppressed allodynia, edema, and hyperemia in the inflamed joints and reduced immune cell infiltration and erosion of joint cartilage [20]. In regards to cell survival, RA-FLS, similarly to carcinoma cells, are more resistant to the changed environment and exhibit an imprinted aggressive phenotype, predisposing them to participate in an inflammatory positive feedback loop in response to cues from the synovial environment [32]. RA-FLS also display invasive capabilities, similar to cancer cells [33]. In vitro, *G. lucidum* polysaccharide peptide (GLPP) significantly inhibited the proliferation of RA-FLS [34]. In addition, certain cells in peripheral blood are also involved in the inflammatory process of RA. In addition, the activation of leukocytes, such as neutrophils, is a major feature of inflammatory diseases [28]. It has been demonstrated that GLP significantly reduces the number of leukocytes in inflamed tissues compared to the inflammatory group [20]. GLP may alleviate inflammatory injury by inhibiting the elevation and aggregation of inflammatory cells.

### 3.2. Secretion of Inflammatory Cytokines

The inflammatory process is usually tightly regulated by pro-inflammatory and anti-inflammatory cytokines. In a chronic inflammatory state, the imbalance between the two types of mediators leaves inflammation unchecked, and this leads to cellular damage [35]. Meanwhile, high levels of pro-inflammatory cytokines and chemokines have been found in the synovial fluid of RA patients [36]. These cytokines and chemokines play important roles in RA-induced synovitis, pannus formation and joint destruction. Macrophage-like synoviocytes produce pro-inflammatory cytokines such as TNF-α, IL-6, IL-1, and FLS are the main source of IL-6, MMP, and prostaglandins [37]. FLS in RA can attract monocytes from the vasculature by secreting chemotactic factors such as monocyte chemoattractant protein-1 (MCP-1) and extend the lifespan of B cells through the production of IL-6, vascular cell adhesion molecule-1 (VCAM-1) [27]. These pro-inflammatory cytokines accumulate more relevant cells in the joint and accelerate their destruction through a paracrine/autocrine pathway [38]. In some inflammatory diseases caused by disorders of the immune system, GLP treatment is reported to inhibit the expression of caspase-3, caspase-9, IL-6, IL-1β and TNF-α and reduces the inflammatory polarization of macrophages [39,40,41,42]. For example, GLP is more potent in inhibiting the level of serum inflammatory cytokines in nude mice, activation of macrophage RAW264.7 and the expression of the inflammatory mediators IL-1β, TNF-α, inducible nitric oxide synthase (iNOS) and cyclooxygenase (COX)-2 in vitro [43]. The activity of MMP-9 could also be inhibited by GLPP in a dose-dependent manner [44]. Moreover, GLP significantly down-regulated pro-inflammatory cytokines, including iNOS, IL-1, and TNF-α, in a dose-dependent manner and up-regulated the anti-inflammation cytokine IL-10 in vivo and in vitro [45,46]. These effects produced by GLP in other inflammatory diseases caused by disorders of the immune system can also be observed in the treatment of RA. Of note, when rheumatoid arthritis synovial fibroblast (RASF) were activated by IL-1β or lipopolysaccharide (LPS), IL-8 and monocyte chemoattractant protein (MCP)-1 production increased many folds, and GLPP significantly suppressed their productions [34]. These results indicated that GLP inhibits the production of inflammatory cytokines and has a protective effect on the preventive treatment of inflammatory response in RA.

### 3.3. Expression of Signaling Pathways

Regarding the mechanism of GLP inhibition of RA inflammation, nuclear transcription factor-kappa B (NF-кB) and mitogen-activated protein kinase (MAPK) pathways may be involved [34,47]. NF-кB and MAPK signaling pathways are classical inflammatory pathways that play important roles in RA joint inflammation [48,49]. Previous studies have confirmed that activation of NF-kB can affect the expression of various inflammatory cytokines such as TNF-α, IL-1β, IL-6, and IL-10 and that activation of NF-kB is also a positive feedback result of inflammatory mediators [50]. Activation and production of NF-κB were found in animal models and in synovial tissue of patients with RA [8]. It has been shown that extracellular signal-regulated kinases (ERKs) of MAPK kinases can be expressed and activated in rheumatoid synovial tissue under appropriate conditions [48,51]. In vascular diseases and inflammation, pretreatment with GLP decreased LPS-induced p65, ERK, p38, JNK, and Akt phosphorylation in human aortic smooth muscle cells [52]. Sang et al. [53] found that GLP treatment inhibited macrophage infiltration and suppressed MAPK (JNK and ERK) activation in RAW264.7. Wang et al. [54] discovered that GLP could down-regulated the expression of iNOS and TNF-α severely inhibited the phosphorylation of IκBα and JNK1/2. All these results indicated that GLP could inhibit the inflammation in RA, at least partially attributed to the blocking of the NF-κB and MAPK pathways.

## 4. Regulation of GLP on Immune Response

RA is a chronic autoimmune disease characterized by immune cell dysfunction and infiltration [55]. Many immune cell clusters, such as DCs, lymphocytes, monocytes/macrophages, and natural killer (NK) cells, are significantly increased in RA and are involved in the pathogenesis of RA [56,57]. DCs serve as powerful antigen-presenting cells (APCs) that continuously present endogenous antigens to activate T cells [58]. B cells act as a source of RF and ACPA and also act as APCs to further promote T cell activation [59]. These contribute to the production of autoantibodies in the joints, thus supporting autoimmunity. Macrophages produce inflammatory cytokines and chemokines that lead to cartilage and bone destruction in RA.

The wide range of pharmacological activities of active ingredients such as *G. lucidum* and its polysaccharides are mainly mediated through its broad immunomodulatory effects [23,60]. In particular, many studies have demonstrated the immunomodulatory effects of GLP in vivo and in vitro [61]. RA is an autoimmune disease, and treatment with immunosuppressive drugs is the generally accepted method. While treated with an immune-promoting substance, it would be contrary to the philosophy of therapy. However, GLP has complicated and two-way regulated functions, with different therapeutic effects in different types of cells and at different functional intensities [62,63]. Many studies proved the immunosuppressive functions of GLP, including regulation of the proliferation and differentiation of APCs such as DCs and inhibition of phagocytosis by mononuclear macrophages and NK cells [60,64]. In addition, GLP has functions in balancing humoral and cellular immunity, such as regulating immunoglobulin production, modulating T and B lymphocyte proliferative responses and cytokine release [65].

### 4.1. Modulatory Effect of GLP on DCs

DCs are the initiators of the immune response, presenting arthritis-associated antigens to T cells and producing cytokines and chemokines that effectively activate T cells, B cells, NK cells and even neutrophils in the inflammatory microenvironment [66]. Cell research has shown that polysaccharides-treated DCs could suppress T cell proliferation with IL-10 production, suggesting that GLP may possess the potential to regulate immune responses [67]. T cells are important players in the immune response to RA [68]. In the induction and development of RA, CD4+ T cell subsets can differentiate to produce a variety of pro-inflammatory cytokines and chemokines. CD8+ T cells might have a suppressive role in the etiology of RA. In the in vivo study, the ratio of CD3+CD4+/CD3+CD8+ T lymphocytes and apoptosis of CD4+ and CD8+ T lymphocytes decreased after being pretreated with *G. lucidum* [69].

### 4.2. Modulatory Effect of GLP on T Cells

Among the numerous CD4+ T cells, Th1 and Th17 are currently considered to be the main T cell subsets involved in the pathogenesis of RA [70]. Th17 was significantly higher in active RA patients than in non-active and healthy comparison groups, and the Th17/Treg ratio was significantly higher than that of healthy individuals, suggesting the existence of Th17/Treg imbalance in RA patients [71,72]. A significant imbalance of Th1 and Th2 cells was also detected in RA patients and animal models [73,74]. The research indicates that if this balance is tilted towards Th1 and Th17 cells in the serum and joint fluid of RA, it can lead to increased levels of inflammatory cytokines such as lL-2 and interferon (IFN)-γ secreted by Th1 cells and cytokines such as IL-17 secreted by Th17 cells. At the same time, the levels of anti-inflammatory cytokines such as IL-4 and IL-10 secreted by Th2 cells and Treg cells were reduced [75]. In this regard, increased IL-17 levels in RA synovial fluid can highlight the importance of IL-17 in the development of RA [76]. Therefore, one of the strategies for treating RA is to reduce the associated inflammation caused by the conversion of Treg cells into Th17 cells. GLP has been shown to have good effects on chronic cerebral hypoperfusion (CCH) by increasing Foxp3 Treg cell levels to increase levels of IL-10 and transforming growth factor (TGF)-β1 and regulate abnormal energy metabolism [77]. Wei et al. found that GLP may suppress the immune responses, including decreasing the secretion of proinflammatory cytokines, such as TNF-α, IL-6, IL-1β, and IL-17A, and decreasing the populations of Th17 cells and NK cells [78]. But now, studies have found that GLP enhances the Th1 response with high levels of IFN-γ and IL-2 and displays low to no impact on IL-4 and IL-10 production. What’s more, GLP induced IL-17 production at different concentration levels [79]. This is detrimental if similar phenomena are observed in RA [62], which requires further research.

### 4.3. Modulatory Effect of GLP on B Cells

B lymphocytes play a key role in the humoral immune response through the production of antibodies against antigens [80]. In synovial tissue, B-cell infiltration, particularly cell infiltration in ectopic lymph, correlates with the severity of RA [80,81]. A portion of the B cells in the synovium differentiate into plasma cells that produce autoantibodies such as RF and ACPA, which stimulate the release of macrophages and FLS cytokines, leading to the development of inflammation [82]. Serum IgG levels are significantly higher in RA patients compared to the general population. This can activate the complement system and cause synovial damage, which has a pathogenic role in RA. The other part differentiates into effector B cells, which produce pro-inflammatory cytokines and promote osteoclastogenesis, leading to bone and joint destruction in RA [83]. It has been demonstrated that GLP limits the homing of lymphocytes from the peripheral blood to the spleen and lymph nodes, suppresses the activation of the complement system and blocks the binding of TNF-α and IFN-γ to their antibodies [41]. Recent experiments by our team showed that a dose of 200 mg/kg/d of GLPP given by gavage for 5 weeks produced good immunomodulatory effects to significantly reduce the number of RF and anti-collagen type II antibodies in CIA rats (submitted).

### 4.4. Modulatory Effect of GLP on Macrophages

Natural immunity also plays an important role in the pathological development of RA [84]. Clinical studies have shown that macrophages are crucial in inducing the progression of inflammation and deterioration of joint destruction in RA [85]. The number of infiltrating macrophages in the lining layer of the synovial membrane correlates with the degree of disease activity and joint erosion in inflamed synovial tissue. Activated macrophages are divided into two groups: M1 macrophages which invade the synovial membrane and secrete pro-inflammatory cytokines (such as IL-1β, IL-6, and TNF-α) and iNOS to participate in the inflammatory response; M2 anti-inflammatory macrophages which reduce inflammation and induce tissue repair by secreting anti-inflammatory cytokines such as IL-10 and TGF [86]. The ratio of M1/M2 macrophages in the synovial fluid of RA patients was significantly higher, and the peritoneal macrophages of AIA rats were mainly M1-polarized [87]. *G. lucidum* significantly reduced TNF-α production by RAW 264.7 and peripheral blood mononuclear cells (PBMCs) from asthma patients [88]. In vitro, GLP can inhibit the production of NO, TNF-α, IL-1β, and IL-6 in RAW 264.7 macrophage stimulated by LPS [89]. Guo et al. [60] found that GLP treatment inhibited macrophage infiltration, down-regulated the expression of IL-1β, iNOS and COX-2, and suppressed LPS-induced activation of inflammatory markers and MAPK (JNK and ERK) in RAW264.7. Besides, Li et al. [90] suggested that GLP could inhibit inflammatory response by inhibiting the M1 inflammatory polarization of macrophages. The bidirectional immunomodulatory effect of GLP would make it superior to other immunosuppressive agents and other herbal medicines used in current autoimmune disease treatment schemes [42]. This property allows GLP to effectively alleviate the severity of disease in RA patients according to their immune status. We hypothesize that different concentrations of GLP may have different effects on individuals with different immune statuses, and pending further cellular and animal experiments are needed to prove this idea.

## 5. Suppression of GLP on Angiogenesis

Angiogenesis and vasculogenesis are common in the development of RA [91]. The proliferation of the intimal layer and recruitment of immune cells (e.g., B cells, T cells, and macrophages) results in an inflammatory and hypoxic environment for the synovium [92]. This can lead to the production of several potent angiogenic factors, including VEGF, fibroblast growth factor (FGF)-2, and hepatocyte growth factor. Hypoxia-inducible factor (HIF)-1, one of the key mediators of RA, induces angiogenesis, promotes FLS migration and cartilage destruction, and inhibits synovial cell apoptosis, eventually may result in the formation of pannus [93].

GLP indeed possesses anti-angiogenic and immune-modulating functions to better protect human tissues and reduce disease progression and health [94,95]. GLPP inhibited angiogenesis by directly inhibiting the cell proliferation of human umbilical cord vascular endothelial cells (HUVEC) [96]. GLPP also directly induced cell death in HUVEC by decreasing the expression of the anti-apoptotic protein Bcl-2 and increasing the expression of the pro-apoptotic protein Bax, which reduced VEGF secretion under hypoxic conditions [97]. In B16F10 melanoma cells, GLP demonstrated the ability to inhibit the production of TGF-β, IL-10 and VEGF [98]. Most importantly, there is scholarly evidence that GLP has the potent anti-inflammatory action to prevent the entry of human aortic smooth muscle cells (HASMCs) into the cell cycle and reduce the intercellular adhesion molecule (ICAM)-1 protein expression in vitro and in vivo [99,100]. Therefore, it is believed that GLP represents a new safe and effective approach to the prevention and treatment of RA by directly inhibiting vascular endothelial cell proliferation and inducing cell death.

## 6. Reduction of GLP on Bone Erosion

As an autoimmune disease characterized by inflammation and bone loss, bone destruction is a typical pathological feature of RA [101]. Insufficient osteoblast-mediated bone formation and excessive osteoclast-regulated bone resorption leading to a disruption of bone homeostasis in RA, causing great pain to the patients [102]. In addition, localized erosion in RA patients is closely related to clinical symptoms and decreased bone mineral density. In a clinical trial, Li et al. [103] have shown that *G. lucidum* can reduce pain in patients with RA. And *G. lucidum* has already been used as a traditional anti-osteoporosis drug for the treatment of bone healing, bone formation and other skeletal disorders [104]. It increases the number of osteoblasts, promotes the maturation of osteoblasts, increases the bone matrix, and matures the bony structures of bone trabeculae [105]. Polysaccharide preparation for subcutaneous injection extracted from Taishan red *G. lucidum* has also shown a protective effect on bone mass in ovariectomized rats [106]. However, bone size and bone mass indicators are evaluated based on subjective scores and should be further consolidated by more specific measurements.

The destruction of articular cartilage is another feature that distinguishes RA from most other connective tissue diseases, and MMPs secreted by FLS contribute greatly to cartilage destruction [101]. In AIA rats, Lam et al. [20] found that the treatment of *G. lucidum* decreased the extent of cell infiltration, tissue proliferation and erosions of joint cartilage.

## 7. Resistance of GLP on Anemia

RA is a multi-systemic disease, and some patients may develop extra-articular manifestations, including anemia, during an episode or progression of RA [107]. Many factors, including impaired IL-6-related iron use, drug-related hepatotoxicity and hematology, and gastrointestinal problems, are causes of rheumatoid anemia [108,109]. The hemoglobin concentration of erythrocytes and red blood cells in the peripheral blood of AIA rats was reduced compared to that of normal rats [110]. GLP selectively binds to bone marrow stromal cells, stimulates the secretion of hematopoietic growth factors, and enhances the clonogenic activities of hematopoietic and stromal cells to promote hematopoiesis in mice [111]. However, the possibility and mechanism of GLP’s anti-anemia properties need to be further investigated.

## 8. Nano-Formulation of GLP

As previously mentioned, the immunosuppressive and anti-inflammatory effects of GLP make it a promising agent for the clinical treatment of RA. Natural polysaccharides have unique properties, including excellent biocompatibility, biodegradability, stability, and essential characteristics of degradable polymers for use as biomaterials. At the same time, polysaccharides have hydrophilic groups, such as hydroxyl groups, which can interact with nanomaterials to prevent agglomeration of nanomaterials [112].

A novel pH-sensitive nanoparticle drug delivery system based on GLP, which had been demonstrated to produce anti-tumor activities, was designed to enable the delivery of methotrexate (MTX) and 10-hydroxycamptothecin (HCPT) to tumor cells, where they could exert synergistic anti-tumor effects, with minimal side effects in vivo [113]. Wang et al. [45] employed a one-step method to prepare selenium nanoparticles (SeNPs) decorated by the water-soluble derivative of GLP. SeNPs-GLP were found to significantly down-regulated pro-inflammatory cytokines, including iNOS, IL-1 and TNF-α in a dose-dependent manner and up-regulated the anti-inflammation cytokine IL-10. All of these results suggest that SeNPs-GLP complexes have anti-inflammatory potential modulating pro-/anti-inflammation cytokine secretion profiles and that the mechanism is partially due to inhibition of activations of NF-κB, JNK1/2 and p38 MAPKs pathways. These nanocarriers offer unique properties, including improved bioavailability and solubility of the drug, reduced systemic adverse effects, prolonged circulation time and the potential for preferential accumulation in the precise target organ, which is expected to lead to improved and efficacious delivery of the administered agent. Thus, nanoparticle-based delivery systems show priority, and these systems may be developed to enhance the therapeutic efficacy of GLP as an anti-RA agent.

## 9. Safety of GLP

GLP is a natural polymer consisting of monosaccharides linked by α and β glycosidic bonds to form main and side chains, with a molecular weight distribution between 103–106 Da. It will have antioxidant activity if the side chain has β-d-Glcp, and it will have immunomodulation activity if the side chain has α-l-Fucp [61]. Generally, GLP is water-soluble and insoluble in alcohol, so methods to isolate GLP usually include aqueous extraction and alcohol precipitation. A few acidic GLPs can be extracted using NaOH solution or Na2CO3 solution at temperatures below 10 °C after hot water extraction. In addition, some scholars have also used enzymes (cellulase and pectinase) to degrade cellulose and pectin in the cell wall or ultrasonic/microwave-assisted extraction of GLP [18,19]. Despite the anti-RA pharmacological effects of GLP in both cellular and animal models of RA (Figure 2), the safety of GLP must be assessed before its clinical application is evaluated. Oral administration of 5000 mg/kg of hot water extract of *G. lucidum* to mice for 30 days did not affect body weight, organ weights or hematological parameters [64]. A GLP-based product named “Ji 731 Injection” has been used clinically for treating polymyositis, dermatitis and myotonic dystrophy in China since 1973. These data suggest that aqueous extracts of *G. lucidum* or GLP are quite safe.

However, because of the hypoglycemic, hypolipidemic, hypotensive, and anticoagulant effects of GLP, to avoid overpowering effects, patients should ask their physicians if they are taking *G. lucidum* or its extract when they are being treated with antidiabetic drugs, hypolipidemic drugs, or anticoagulants at the same time. In addition, because of its sedative and hypnotic effects, *G. lucidum* may also potentiate the effects of sedative drugs [114,115]. GLP also has an antibacterial effect, increasing the activity of certain antibiotics such as tetracycline and cefazolin [116,117]. Therefore, GLP should also be given with great care for patients of RA who are treated with other drugs for the combined diseases.

## 10. Conclusions and Future Directions

RA is a chronic, inflammatory and autoimmune disease characterized by inflammation, autoimmune dysfunction, and cartilage and bone destruction. In this review, we summarize the available reports on the protective effects of GLP against RA in terms of anti-inflammatory, immunomodulatory, anti-angiogenic and osteoprotective effects, as shown in Table 1. Firstly, GLP inhibits RASF proliferation and migration, modulates pro- and anti-inflammatory cytokines and reduces synovial inflammation. Secondly, GLP regulates the proliferation and differentiation of APCs such as DCs and inhibition of phagocytosis by mononuclear macrophages and NK cells and regulates the ratio of M1, M2, and M1/M2-related inflammatory cytokines. In addition, GLP has functions in balancing humoral and cellular immunity, such as regulating immunoglobulin production, modulating T and B lymphocyte proliferative responses and cytokine release, and exhibiting immunomodulatory effects. Thirdly, GLP inhibits angiogenesis through direct inhibition of vascular endothelial cell proliferation and induction of cell death and indirect inhibition of VEGF production in tumor cells. Finally, GLP can inhibit the production of MMPs and promote osteoblast formation, exerting protective effects on bone and articular cartilage. In addition, the text also briefly describes the anti-anemic effects of GLP.

All of the above results, as well as the available literature, suggest that GLP may be a promising agent for the treatment of RA. Many of the mechanisms of GLP for the treatment of RA remain unknown. Firstly, most of the knowledge of the anti-arthritic mechanisms of GLP comes from other cellular models; therefore, there is a need for more research in animal and cellular models of RA. Secondly, more experiments are needed to confirm the anti-arthritic mechanisms of GLP, particularly the anti-inflammatory and skeletal protective pharmacological effects. Thirdly, further clinical trials are needed to investigate the safety and efficacy of GLP. Although these studies have a long way to go to guide the clinical use of GLP, it is hoped that with improved formulation, GLP will become a new and effective agent for RA.

## Figures and Tables

**Figure 1 molecules-28-02483-f001:**
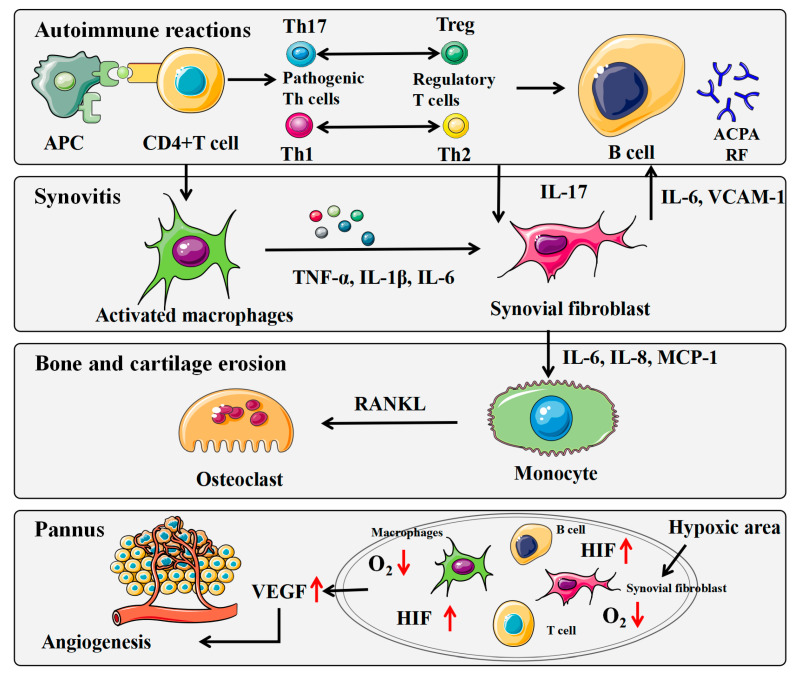
The pathological mechanism of rheumatoid arthritis (RA). APC: antigen-presenting cell; DC: dendritic cell; ACPA: anti-citrullinated protein/peptide antibodies; RF: rheumatoid factor; TNF: tumor necrosis factor; IL: interleukin; VCAM-1: vascular cell adhesion molecule-1; MCP-1: monocyte chemoattractant protein-1; RANKL: receptor activator of nuclear factor-κB ligand; VEGF: vascular endothelial growth factor; HIF: hypoxia-inducible factor; Red arrow mean up or down regulation.

**Figure 2 molecules-28-02483-f002:**
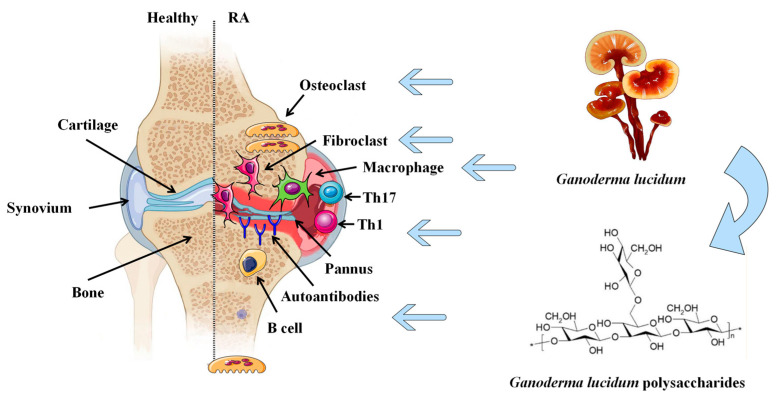
Potential anti-RA effects of GLP. GLP can inhibit the proliferation and migration of fibroblasts and regulate inflammatory cytokines. It exhibits immunomodulatory effects by regulating the proliferation and differentiation of T and B lymphocytes. GLP attenuates pannus formation by inhibiting the proliferation of vascular endothelial cells and promoting the formation of osteoblasts, exerting a protective effect on bone and joint cartilage. RA: rheumatoid arthritis.

**Table 1 molecules-28-02483-t001:** The role of GLP in different experimental models/cell types.

Years	Authors	Doses of GLP	Experimental Model/Cell Type	Effects of GLP
2004	Cao et al. [96]	1, 10 and 100 mg/L	HUVECs	GLPP inhibits angiogenesis by directly inhibiting the cell proliferation of HUVEC.
2006	Cao et al. [97]	10 and 100 μg/mL	HUVECs	GLPP induced cell death in HUVEC by decreasing the expression of Bcl-2 and increasing the expression of Bax, which reduced VEGF secretion under hypoxic conditions.
2007	Ho et al. [33]	50, 125 and 250 μg/mL	RASF	GLPP significantly inhibited the proliferation of RASF and suppressed their production by inhibiting the NF-κB pathway.
2010	Lin et al. [99]	0.25 and 0.5 μg/mL	HASMCs	GLP suppressed LPS-induced ICAM-1 mRNA and protein expression and reduced the binding of human monocytes to LPS-stimulated HASMCs.
2012	Wang et al. [100]	100 mg/kg	male C57BL/6 J mice	GLP decreased the neointimal area in vivo.
2014	Wang et al. [53]	5, 20, 50 and 100 μg/mL	RAW264.7	GLP down-regulated the expression of iNOS and TNF-α and severely inhibited the phosphorylation of IκBα and JNK1/2.
2014	Zhong et al. [107]	50 mg/kg	Male Kunming mice	GLP could increase antioxidant enzyme activities and decrease the MDA levels in the skeletal muscle of mice.
2018	Wei et al. [77]	100 mg/kg	male C57BL/6 J mice	GLP decreased the secretion of proinflammatory cytokines, such as TNF-α, IL-6, IL-1β, and IL-17, and decreased the populations of Th17 cells and NK cells.
2021	Sang et al. [52]	100 and 300 mg/kg	male C57BL/6 J mice	GLP inhibited macrophage infiltration and suppressed MAPK (JNK and ERK) activation.
2021	Guo et al. [59]	0.2, 0.4, 0.8 and 1.6 mg/mL	RAW264.7	GLP inhibited LPS-induced inflammation markers and MAPK (JNK and ERK) activation in macrophage RAW264.7.
2022	Fang et al. [42]	0.625, 1.25 and 2.5 mg/mL	RAW264.7	GLP inhibited the activation of macrophage RAW264.7 and the expression of the inflammatory mediators IL-1β, TNF-α, iNOS, and COX-2.
2022	Jia et al. [89]	1, 10, and 100 μg/mL	RAW264.7	GLP can inhibit the production of NO, TNF-α, IL-1β, and IL-6 in RAW 264.7 macrophage stimulated by LPS.

GLP: *Ganoderma lucidum* polysaccharide; GLPP: *Ganoderma lucidum* polysaccharide peptide; RASF: rheumatoid arthritis synovial fibroblasts; NF-κB: nuclear transcription factor-kappa B; IL: interleukin; TNF: tumor necrosis factor; MAPK: mitogen-activated protein kinase; iNOS: inducible nitric oxide synthase; COX: cyclooxygenase; LPS: lipopolysaccharide; HUVEC: human umbilical vascular endothelial cell; VEGF: vascular endothelial growth factor; ICAM: intercellular adhesion molecule; HASMC: human aortic smooth muscle cells.

## Data Availability

Not applicable.

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
