# Peer review of "Potential Anti-Rheumatoid Arthritis Activities and Mechanisms of Ganoderma lucidum Polysaccharides"

_molecules, 2023, doi:10.3390/molecules28062483_

Round 1

Reviewer 1 Report

This review could provide an important insight into the relationship between RA and GLP. Overall this review covers well the novel and interesting issues and therapeutic use of GLP in RA and is well written. However, there several concerns that should be revised as follows;

In abstract, proper English editing process is needed such as (bone and articular cartilage protective effects-> protective effects of bone and articular cartilage.)

In section from 3-7, authors represent various role of GLP. Change the subtitles to clarify the effect of GLP.

Abbreviation of word must be represented in the beginning of description (e.g. line 47, 53). Check the location of abbreviation throughout the manuscript.

In figure 1 and section 3, pathological mechanism of RA should be modified. Because synovial fibroblast becomes not only cytokine-accepting cells but also cytokines-releasing cells. Add this point in the text and references.

Describe the molecular mechanism of GLP and represent its illustrated mechanism.

In section 3 and 4, these sections are too long and it is hard to follow. Use the subtitle and sort the section.

Author Response

We are very grateful for your comments regarding our manuscript "The potential anti-rheumatoid arthritis activities and mechanisms of Ganoderma lucidum polysaccharides" (ID: molecules-2236559). We have reproduced the comments verbatim and corrected the manuscript carefully which we hope that the current version will be satisfactory and acceptable for publication in Molecules.

The revised manuscript has been uploaded with the point-to-point response letter, and all changes have been used by “Track Changes” function.

            Thanks again for your efforts and time in handling our manuscript and we are looking forward to your comments.

Reviewer: 1

Comments to the Author: This review could provide an important insight into the relationship between RA and GLP. Overall this review covers well the novel and interesting issues and therapeutic use of GLP in RA and is well written. However, there several concerns that should be revised as follows.

Comment 1: In abstract, proper English editing process is needed such as (bone and articular cartilage protective effects-> protective effects of bone and articular cartilage.)

Reply: Thank you for this insightful comment. We have changed the “bone and articular cartilage protective effects” into “protective effects of bone and articular cartilage” (Line 26, 402).

Comment 2: In section from 3-7, authors represent various role of GLP. Change the subtitles to clarify the effect of GLP.

Reply: Thank you for your helpful comments. We have changed the subtitles to clarify the effect of GLP in section from 3-7.

Comment 3: Abbreviation of word must be represented in the beginning of description (e.g. line 47, 53). Check the location of abbreviation throughout the manuscript.

Reply: Thank you for pointing out this problem. We have checked the location of abbreviation throughout the manuscript and corrected this mistake in the manuscript (Line 47, 79, 80, 83, 85, 178, 195, 264).

Comment 4: In figure 1 and section 3, pathological mechanism of RA should be modified. Because synovial fibroblast becomes not only cytokine-accepting cells but also cytokines-releasing cells. Add this point in the text and references.

Reply: Thank you for pointing out the problem. We have added this point in the manuscript (section 3.2.).

Comment 5: Describe the molecular mechanism of GLP and represent its illustrated mechanism.

Reply: Thank you for pointing out the problem. We have described the molecular mechanism of GLP and represent its illustrated mechanism in figure legend (Figure 2).

Comment 6: In section 3 and 4, these sections are too long and it is hard to follow. Use the subtitle and sort the section.

Reply: Thank you for pointing out the problem. We have added the subtitle in the manuscript in section 3 and 4.

The above is our reply to the reviewer's comments. And these changes will not influence the content and framework of the manuscript. We would like to thank the reviewer again for your efforts and time reviewing and handling our manuscript.

Reviewer 2 Report

The full name of MMPs should be added.

Author Response

We are very grateful for your comments regarding our manuscript "The potential anti-rheumatoid arthritis activities and mechanisms of Ganoderma lucidum polysaccharides" (ID: molecules-2236559). We have reproduced the comments verbatim and corrected the manuscript carefully which we hope that the current version will be satisfactory and acceptable for publication in Molecules.

The revised manuscript has been uploaded with the point-to-point response letter, and all changes have been used by “Track Changes” function.

            Thanks again for your efforts and time in handling our manuscript and we are looking forward to your comments.

Reviewer: 2

Comment 1: The full name of MMPs should be added.

Reply: Thank you for pointing out the problem. We have added the full name of MMPs in the manuscript (Abstract).

The above is our reply to the reviewer's comments. And these changes will not influence the content and framework of the manuscript. We would like to thank the reviewers again for your efforts and time reviewing and handling our manuscript.

Reviewer 3 Report

This article provides relevant literature with a good overview of the etiology and molecular mechanisms of rheumatoid arthritis (RA), and the protective effects of Ganoderma lucidum polysaccharide (GLP) on RA-related damage, including anti-inflammatory, immunomodulatory, anti-angiogenic, anti-anemic and bone-protective effects.

Specific comments:

1.    I suggest that "The" in the first word of the title can be deleted.

2.    It is recommended that the authors give a brief overview of the preparation methods and chemical properties of "GLP" and "GLPP" in this article, so that readers can understand the basic properties of "GLP" and "GLPP".

3.    Line 145: The word “NF-кb” should be “NF-кB”.

4.    Figure 2: The words “Ganoderma lucidum” should be italicized (appeared twice in this figure).

Author Response

We are very grateful for your comments regarding our manuscript "The potential anti-rheumatoid arthritis activities and mechanisms of Ganoderma lucidum polysaccharides" (ID: molecules-2236559). We have reproduced the comments verbatim and corrected the manuscript carefully which we hope that the current version will be satisfactory and acceptable for publication in Molecules.

The revised manuscript has been uploaded with the point-to-point response letter, and all changes have been used by “Track Changes” function.

            Thanks again for your efforts and time in handling our manuscript and we are looking forward to your comments.

Reviewer: 3

Comment 1:  I suggest that "The" in the first word of the title can be deleted.

Reply: Thank you for pointing out the problem. We have deleted "The" in the first word of the title.

Comment 2: It is recommended that the authors give a brief overview of the preparation methods and chemical properties of "GLP" and "GLPP" in this article, so that readers can understand the basic properties of "GLP" and "GLPP".

Reply: Thank you for your helpful comments. We have added the preparation methods and chemical properties of "GLP" in the manuscript (Section 9.).

Comment 3: Line 145: The word “NF-кb” should be “NF-кB”.

Reply: Thank you for pointing out the problem. We have changed this mistake in the manuscript (section 3.3).

Comment 4: Figure 2: The words “Ganoderma lucidum” should be italicized (appeared twice in this figure).

Reply: Thank you for pointing out the problem. We have changed this mistake in the Figure 2.

The above is our reply to the reviewer's comments. And these changes will not influence the content and framework of the manuscript. We would like to thank the reviewer again for your efforts and time reviewing and handling our manuscript.